# Investigations into the biosynthesis of stieleriacines and related *N*-acyl tyrosines by comparative genomics, knock-out studies and total synthesis of *epi*-stieleriacine C
Maria Sauer[1], Myriel Staack [ID][2], Sven Balluff[1], Christian Jogler[2,3], Nicolai Kallscheuer [ID][2][✉] & Christine Beemelmanns [ID][1,4][✉]

*N*-acyl tyrosines, a prominent class of *N*-acyl amino acid biomolecules, are produced by selected species in at least three bacterial phyla: *Pseudomonadota*, *Actinomycetota* and *Planctomycetota*. Long-chain *N*-acyl tyrosines with a characteristic 2,3-dehydrotyrosine core structure and additional taxon-specific chemical modifications were previously reported under the names thalassotalic acids, kyonggic acids and stieleriacines. However, the underlying pathway for their biosynthesis in the different bacterial taxa remains largely unexplored. Here, we focused on the identification of biosynthetic enzymes in the two known stieleriacine-producing planctomycetal strains of the eponymous genus *Stieleria*. Comparative genome analyses of stieleriacine-, thalassotalic acid- and kyonggic acid producers suggest a common pathway for *N*-acyl dehydrotyrosine biosynthesis based on conserved genes encoding a putative adenylyltransferase/cyclase, nitroreductase and the hallmark protein *N*-acyl amino acid synthase (NasY). The targeted deletion of three predicted *nasY* genes in *Stieleria neptunia* indicates that one of the three encoded enzymes predominantly produces stieleriacines. We also confirmed the absolute structure of stieleriacine C by synthesis of its epimer and structural derivatives, which serve as the basis for the future investigation of the biological function of *N*-acyl tyrosines.

*N*-acyl amino acids (NAAs) consist of two components, a fatty acid and an amino acid linked by an amide bond, and are widely distributed across various organisms, tissues, and bacterial membranes[1–3]. While simple in their structural components, their structural variability in terms of saturation degree and carbon number renders this class of signaling molecule highly diverse exhibiting species- or context-specific biological functions[4].

In humans, NAAs influence, e.g., immune homeostasis, building of fat mass levels, regulation of energy expenditure related to obesity, and affect other processes such as pain, memory, and insulin levels[5]. One of the structurally most simple NAAs are glycine lipids (GlyLs), which were

initially characterized as cytolipin in the gliding bacterium *Cytophaga johnsonae* (recommended name *Flavobacterium johnsoniae*) (Fig. 1)[6]. Since their discovery, GlyLs and their related glycine-serine dipeptido-lipids, known as flavolipins (FLs), have been identified in numerous members of the phylum *Bacteroidota*, including species associated with the gut and oral microbiomes[7–9]. NAA-derived products from human commensal bacteria are hypothesized to influence G-protein-coupled receptors via chemical mimicry of eukaryotic metabolites[10]. They have also been shown to affect host behavior, notably enhancing motivation for physical activity through fatty acid amide-dependent activation of the endocannabinoid receptor CB1 and subsequent stimulation of TRPV1 (Transient Receptor Potential

[1]Department Antiinfectives from Microbiota, Helmholtz Institute for Pharmaceutical Research Saarland (HIPS), Saarbrücken, Germany. [2]Department of Microbial Interactions, Institute for Microbiology, Friedrich Schiller University, Jena, Germany. [3]Cluster of Excellence Balance of the Microverse, Friedrich Schiller University, Jena, Germany. [4]Saarland University, Saarbrücken, Saarbrücken, Germany. [✉]e-mail: nicolai.kallscheuer@uni-jena.de; christine.beemelmanns@helmholtz-hips.de

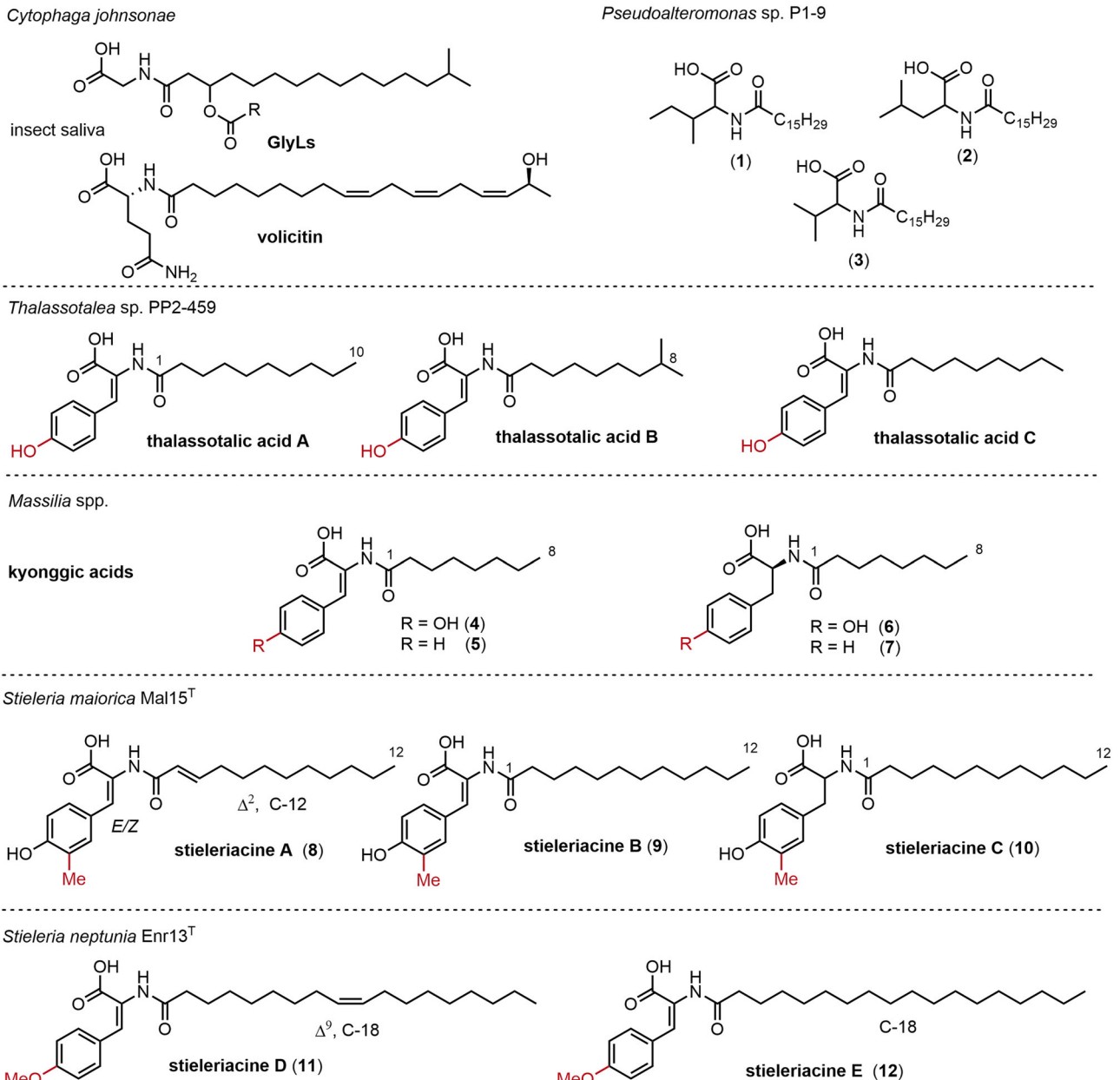

**Fig. 1 | Chemical structures of naturally occurring bacterial *N*-acyl amino acids.**

Vanilloid 1)-sensory neurons in the host periphery[11]. Another example of an acylated polar amino acid is volicitin, which consists of a glutamate head group that is connected to a triple-unsaturated fatty acid via an amide linkage and is found in insect saliva and responsible for induction of chemical defenses in plants upon insect grazing[12].

Several studies showed that marine bacteria are an important source of structural variants of NAAs[13]. Aliphatic NAAs (**1**-**3**) isolated from the marine γ-proteobacterium *Pseudoalteromonas* sp. P1-9 were found to have moderate antimicrobial activity[14]. Tyrosine-derived NAAs, termed thalassotalic acids, have been isolated from the γ-proteobacterium *Thalassotalea* sp. PP2-459[15]. A related family of *N*-acylated tyrosines, called stieleriacines A-C, has been identified in the planctomycete *Stieleria maiorica* Mal15[T]. Stieleriacines D and E, that differ from stieleriacines A-C in fatty acid chain length and tyrosine aromatic ring methylation patterns, have been isolated from *Stieleria neptunia* Enr13[T] (Fig. 1)[16,17]. In *S. maiorica* Mal15[T], stieleriacines appear to play a role in microbial interactions by reducing its lag phase and modulating biofilm formation of competing

bacteria; thereby influencing microbial community composition strategically. Beyond their ecological role, stieleriacines share notable structural features with NAAs identified through the heterologous expression of environmental DNA in *Escherichia coli*[18,19]. Only recently, a third group of structurally related tyrosine and phenylalanine derivatives named kyonggic acids (**4**-**7**) have been isolated from *Massilia* spp.[20], while kyonggic acid congeners were described before from heterologous expression of environmental DNA[21,22]. The three compound families, thalassotalic acids, stieleriacines and kyonggic acids are structurally closely related, but show key differences in acyl chain length and aromatic ring substitution pattern[23]. In addition, thalassotalic acids A-C, stieleriacines A, B, D and E as well as kyonggic acids **4** and **5** exhibit a double bond at the 2,3-position of the tyrosine moiety and thus belong to the 2,3-dehydroamino acids often found in ribosomally and non-ribosomally-synthesized peptides[24]. In contrast, kyonggic acids **6** and **7** and stieleriacine C represent saturated derivatives, and could be either representatives of biosynthetic precursors or side products thereof.

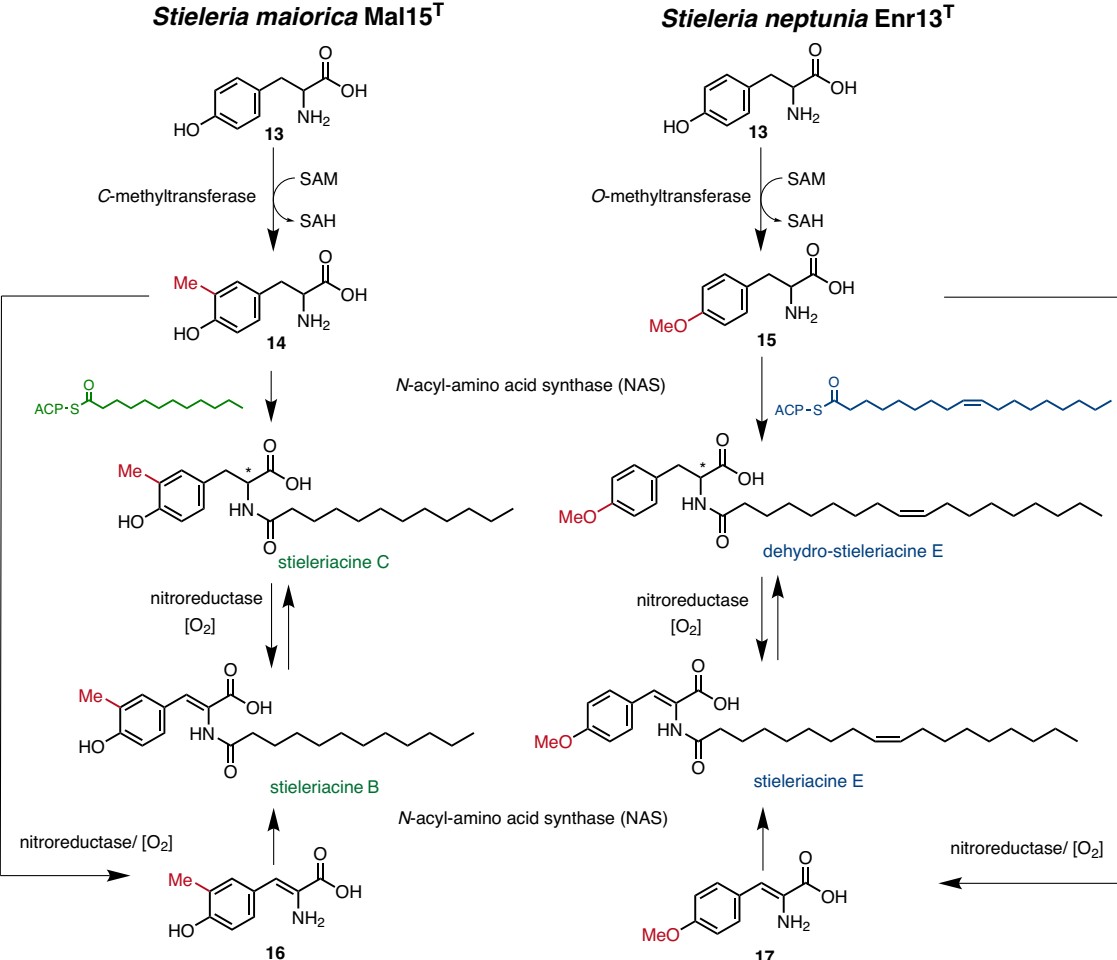

**Fig. 2 |** Proposed biosynthetic transformations involved in the biosynthesis of *O*- and *C*-methylated stieleriacines in *S. maiorica* Mal15[T] and *S. neptunia* Enr13[T] (SAM *S*-adenosyl methionine, SAH *S*-adenosyl-L-homocysteine).

NAAs are likely biosynthesized from ligation of an acyl carrier protein (ACP)-activated fatty acid thioester (acyl-ACP) presumably derived directly from primary metabolism, and an amino acid, which can be either a canonical or non-canonical amino acid harboring further modifications[2]. Dedicated *N*-acyl amino acid synthases (NASs) catalyze the ligation of the fatty acid to the amino group of the amino acid derivative, forming an amide bond in the resulting product[25,26]. As an example, in *Bacteroides thetaiotaomicron*, the biosynthesis of GlyL begins with the *N*-acylation of glycine with a primary β-hydroxy fatty acid through the *N*-acyltransferase encoded by the *glsB* gene[27]. This reaction produces a mono-acylated amine, such as *N*-acyl-β-hydroxy-palmitoyl glycine (commendamide), which then undergoes *O*-acylation (esterification) with a secondary fatty acid, catalyzed by an *O*-acyltransferase encoded by the *glsA* gene, yielding a mature diacylated amino acid lipid. While the biosynthesis of many NAAs has been studied, the underlying biosynthetic pathways for the modified NAAs such as thalassotalic acids (*Thalassotalea* sp.; phylum *Pseudomonadota*), kyonggic acids (*Massilia* spp.; phylum *Pseudomonadota*), and stieleriacines (*Stieleria* spp.; phylum *Planctomycetota*) remain to be elucidated.

We initiated our study with a comparative genomics approach to search for biosynthetic gene clusters (BGCs) potentially involved in the formation of the three groups of *N*-acyl tyrosines. The analysis was guided by genes encoding putative NASs, the hallmark proteins for *N*-acyl amino acid biosynthesis. The computational analysis was complemented by wet lab studies that included the construction of knock-out mutants in the stieleriacine D- and E-producing *S. neptunia* Enr13[T] and by chemical total synthesis of *epi*-stieleriacine C and non-natural derivatives.

## Results and discussion
### Comparative genomics of thalassotalic acid, stieleriacine and kyonggic acid producers

For the identification of putative *N*-acyl phenylalanine/tyrosine-associated BGCs, the genomes of the thalassotalic acids-producer *Thalassotalea* sp. PP2-459 (NCBI acc. no. GCF_001913705.1), the two stieleriacine-producing planctomycetal strains *S. neptunia* Enr13[T] (acc. no. GCF_007754155.1) and *S. maiorica* Mal15[T] (acc. no. GCF_008035925.1) as well as the kyonggic acids producer *Massilia kyonggiensis* TSA1[T] ( = JCM 19189) (acc. no. GCF_024756235.1) were comparatively analyzed (Supplementary Table 1).

First, genomes were mined for putative NAS-encoding homologous gene sequences, while in a second step genes coding for modifying enzymes, such as *O*- or *C*-methyltransferases and oxidoreductases likely responsible for introduction of the double bond (Fig. 2), were taken into consideration. *S. maiorica* Mal15[T] that produces stieleriacines A-C bearing a *meta-C*-methylation on the aromatic ring was expected to encode a dedicated *C*-methyltransferase, while *S. neptunia* Enr13[T] that produces stieleriacines D and E was hypothesized to encode an *O*-methyltransferase homolog. For comparison, a close relative of *M. kyonggiensis*, *Massilia umbonata* LP01[T] (acc. no. GCF_005280315.1) was included since it produces the saturated kyonggic acid derivatives **6** and **7** that resemble stieleriacine C (**10**)[20] and thus the producer strain was expected to lack the hypothesized nitro-/oxidoreductase involved in the double bond formation.

Our comparative analysis of six genomes revealed several clusters, in which either one, three or four putative NAS-encoding genes are co-located

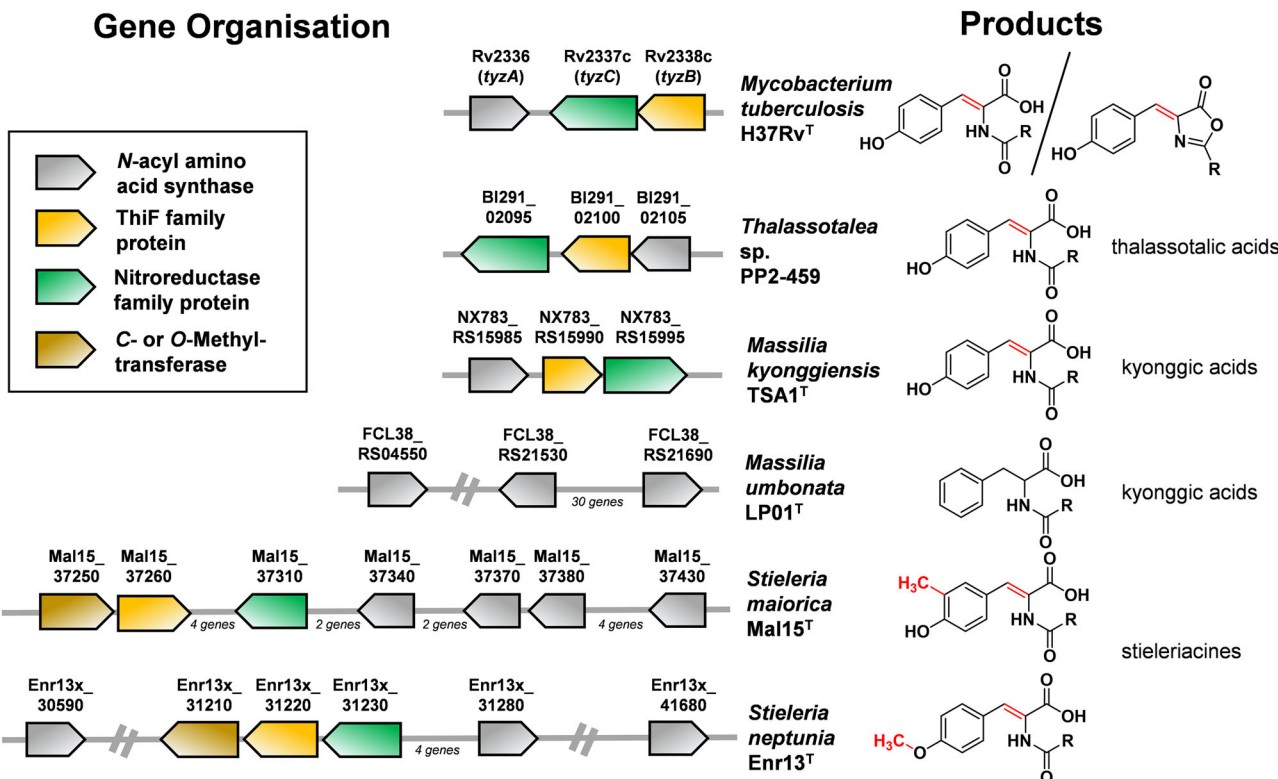

**Fig. 3 | Genetic organization of *N*-acyl amino acid synthase-encoding genes in the analyzed genomes and structure of isolated *N*-acyl (dehydro)tyrosine or phenylalanine derivatives.** Only genes putatively involved in the biosynthesis of tyrazolones, thalassotalic acids, kyonggic acids, and stieleriacines are shown. The analysis is based on the RefSeq-annotated genomes available from NCBI (locus tags are provided on the arrows). For simplicity, the length and variability of the acyl chain (R) is omitted.

within a certain proximity with putative tailoring genes (Fig. 3) based on our biosynthetic proposal (Fig. 2). The genomes of *M. kyonggiensis* (producer of kyonggic acids) and *Thalassotalea* sp. PP2-459 (producer of thalassotalic acids) each contain a gene cluster comprising a single putative NAS-encoding gene co-located with a gene encoding a ThiF family protein (RefSeq locus tags NX783_RS15590 and Bl291_02100, respectively), as well as a gene encoding a putative nitroreductase family protein (NX783_RS15595 and Bl291_02095) (Fig. 2). The nitroreductase family proteins are known to catalyze desaturation reactions, suggesting a functional link to the encoded NAS. This potential relationship is further supported by the operon-like organization of the genes, indicated by short intergenic regions of less than 10 base pairs or even overlapping open reading frames in both, *Thalassotalea* sp. PP2-459 and *M. kyonggiensis*. The encoded nitroreductase family proteins show similarity to TyzC (Uniprot entry P95233), which catalyzes the $O_2$-dependent desaturation of *N*-acyl tyrosine in the $C_{12:0}$-tyrazolone biosynthesis pathway of *Mycobacterium tuberculosis*[28]. The detected ThiF-like adenylyltransferase/cyclase family proteins show similarity to TyzB (Uniprot entry P95234), which catalyzes the ATP-dependent cyclization of *N*-acyl tyrosine/2,3-dehydrotyrosine yielding the heterocyclic tyrazolones in *M. tuberculosis*[28].

Homologous genes encoding ThiF- and nitroreductase family proteins were also identified within the predicted BGCs of the two *Stieleria* species known to produce stieleriacines (Fig. 3). In contrast, these two genes are absent in the candidate BGCs of *M. umbonata* (only producing saturated kyonggic acids), reinforcing the association between their presence and the biosynthesis of compounds containing the 2,3-double bond in the tyrosine moiety via a $O_2$-dependent desaturation. However, the detailed role and biochemistry of ThiF family proteins in the biosynthesis of stieleriacines, kyonggic acids or thalassotalic acids remains enigmatic as yet no acyl-oxazolone derivatives of these three compound families (as present in *M. tuberculosis*) were observed.

A closer examination of possible BGC candidates in *Stieleria* spp. revealed one region in *S. neptunia* (*O*-methylated stieleriacines) that entails one NAS-encoding gene homolog in proximity to three genes encoding modifying enzymes: a methyltransferase (locus Enr13x_31210), a ThiF family protein (locus Enr13x_31220) and a nitroreductase family protein (locus Enr13x_31230). In contrast, the genome of *S. maiorica* (*C*-methylated stieleriacines) harbors a longer region that entails four putative NAS genes within the proximity of the necessary accessory genes, encoding again a methyltransferase (locus Mal15_37250), a ThiF-family protein (locus Mal15_37260), and a nitroreductase family protein (locus Mal15_37310).

To further investigate the biosynthesis of stieleriacines in *Stieleria* spp., we analyzed the potential substrate specificity of the identified putative biosynthetic enzymes by first constructing a phylogenetic tree based on NAS protein sequences from the genomes under study. However, the resulting clustering primarily reflected the phylogenetic relationships among the strains rather than functional differences, suggesting that redundant or overlapping activities in strains containing multiple NAS enzymes could not be ruled out at this stage (Fig. 4A). We therefore focused on the distinguishing feature of *C*- versus *O*-methylation in stieleriacines. To investigate this, a second phylogenetic tree was constructed using sequences of characterized *C*- and *O*-methyltransferases (Fig. 4B), along with those located in close proximity to the identified *nasY* genes. For *S. maiorica*, the methyltransferase protein sequence clustered with the tyrosine-*C*3-methyltransferase SfmM2 (saframycin BGC) of *Streptomyces lavendulae*[29], and is due to the related functions hypothesized to be involved in the *C*-methylation of the stieleriacine precursor (Fig. 2).

In case of *S. neptunia*, the candidate methyltransferase encoded in the putative BGC (locus Enr13x_31210) clustered unexpectedly together with the ubiquinone *C*-methyltransferase UbiE of *E. coli* and thus was considered a less likely candidate for the biosynthesis of *O*-methylated stieleriacines. A broader manual targeted genome analysis uncovered an

**Fig. 4 | Maximum likelihood phylogenetic trees based on protein sequences. A** Tree showing the clustering of candidate *N*-acyl amino acid synthases from *Stieleria* spp., *Thalassotalea* sp. PP2-459 *and Massilia* spp. Since not all proteins were listed in the UniProt database, the NCBI protein accession numbers are shown in brackets instead. **B** Tree showing the clustering of candidate methyltransferases from *S. neptunia* and *S. maiorica* with characterized methyltransferases (in bold). The UniProt entry numbers for all analyzed sequences are shown in brackets. Bootstrap values from 1000 re-samplings are shown on the branches (in %). The tree scale indicates the number of substitutions per amino acid position.

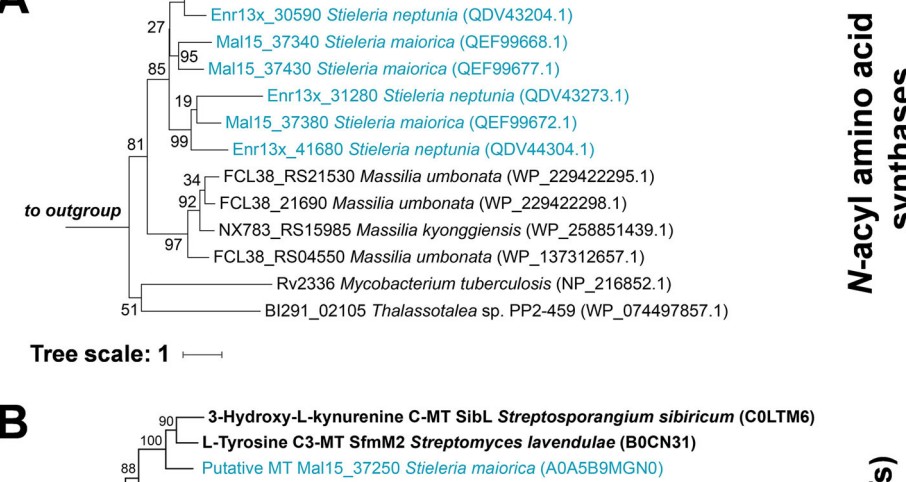

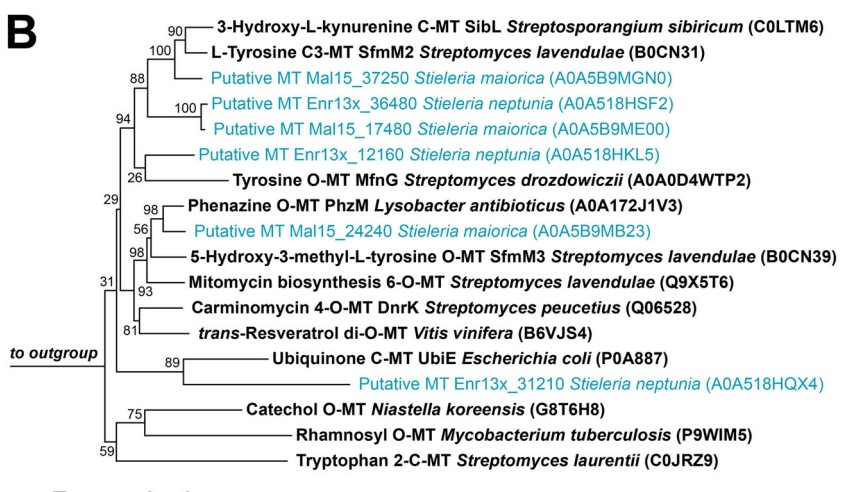

additional alternative candidate (locus Enr13x_12160), which clustered next to a homolog of a tyrosine *O*-methyltransferase MfnG (marformycin BGC, *Streptomyces drozdowiczii*)[23], and was thus considered a more likely candidate for the *O*-methylation of the stieleriacine precursor in *S. neptunia*.

Overall, our findings are consistent with previous reports showing that, in the phylum *Planctomycetota*, biosynthetic pathways for secondary metabolites are often not encoded by clustered genes. This limits the predictive power of current algorithms, which are primarily trained on well-characterized clusters from distantly related taxa, and thus requires manual curation of datasets and in-depth molecular biological studies for validation.

## Analysis of stieleriacine production in wild type and deletion mutants of *S. neptunia*

Given that the most significant variable in the biosynthetic pathway was the localization and nature of the putative NAS enzymes, we sought to first spearhead the identification of the most likely functional candidate through gene knockout experiments in the genetically tractable *S. neptunia*. The type strain Enr13[T] encodes three putative NAS candidates referred to as NasY1, 2 and 3 in the following sections. Single-gene deletion mutants were generated using a double homologous recombination approach. This method replaced the coding region of the targeted *nasY* gene with a chloramphenicol resistance gene, which served as a positive selection marker. Mutants were named *S. neptunia* Δ*nasY1* (deletion of locus Enr13x_30590), *S. neptunia* Δ*nasY2* (locus Enr13x_31280; in the BGC) and *S. neptunia* Δ*nasY3* (locus Enr13x_41680). The deletion mutants were subsequently analyzed for stieleriacine production using a semi-targeted high-resolution tandem mass spectrometry (MS/MS) analysis and comparison to wild-type production[30]. Notably, in addition six previously undetected stieleriacine derivatives were identified in the culture broth of wild-type *S. neptunia*, which clustered within the molecular network of stieleriacines (Fig. 5A). Based on a detailed

analysis of their HRMS/MS fragmentation patterns, we were able to propose preliminary structures for these detectable derivatives; however, precise determination of their absolute structures was not yet possible due to their low production levels.

Extracted ion chromatogram analysis targeting the molecular features of all detectable stieleriacines in the Δ*nasY* mutants revealed that Δ*nasY2* and Δ*nasY3* maintained production levels comparable to the wild type (Fig. 5A, B). In contrast, the Δ*nasY1* mutant showed a significantly reduced stieleriacine production, with levels of all derivatives approaching the detection limit (Supplementary Figs. 1–8). As we expect that residual production observed in Δ*nasY1* may be attributed to genomic redundancy due to the presence of multiple *nasY* copies, these findings led to the conclusion that NasY1 (locus Enr13x_30590) is the primary enzyme responsible for stieleriacine biosynthesis. These results suggest, once again, that the co-localization of biosynthetic genes in *Planctomycetota* appears to be more the exception than the rule and future knock-out studies for each individual gene candidate of a biosynthetic pathway using the herein established procedure will be required to fully elucidate the biosynthetic principle in *Planctomycetota*.

## Synthesis of *N*-acyl tyrosine derivatives and *epi*-stieleriacine C

To complement future biosynthetic studies on the stieleriacine/kyonggic acid biosynthesis and circumvent the labor-intensive process of isolation from natural sources, we also decided to synthesize a small, yet representative library of NAA derivatives featuring the stieleriacine core structure, including the first synthesis of stieleriacine derivative C as a reliable substitute for the future quantification of stieleriacines[31]. As starting materials for the amino acid building blocks, commercially available H-Tyr-OMe*HCl, H-Tyr(Me)-OMe*HCl, and synthesized H-Tyr(3-Me)-OMe*HCl were selected. The use of methyl esters was intended to prevent

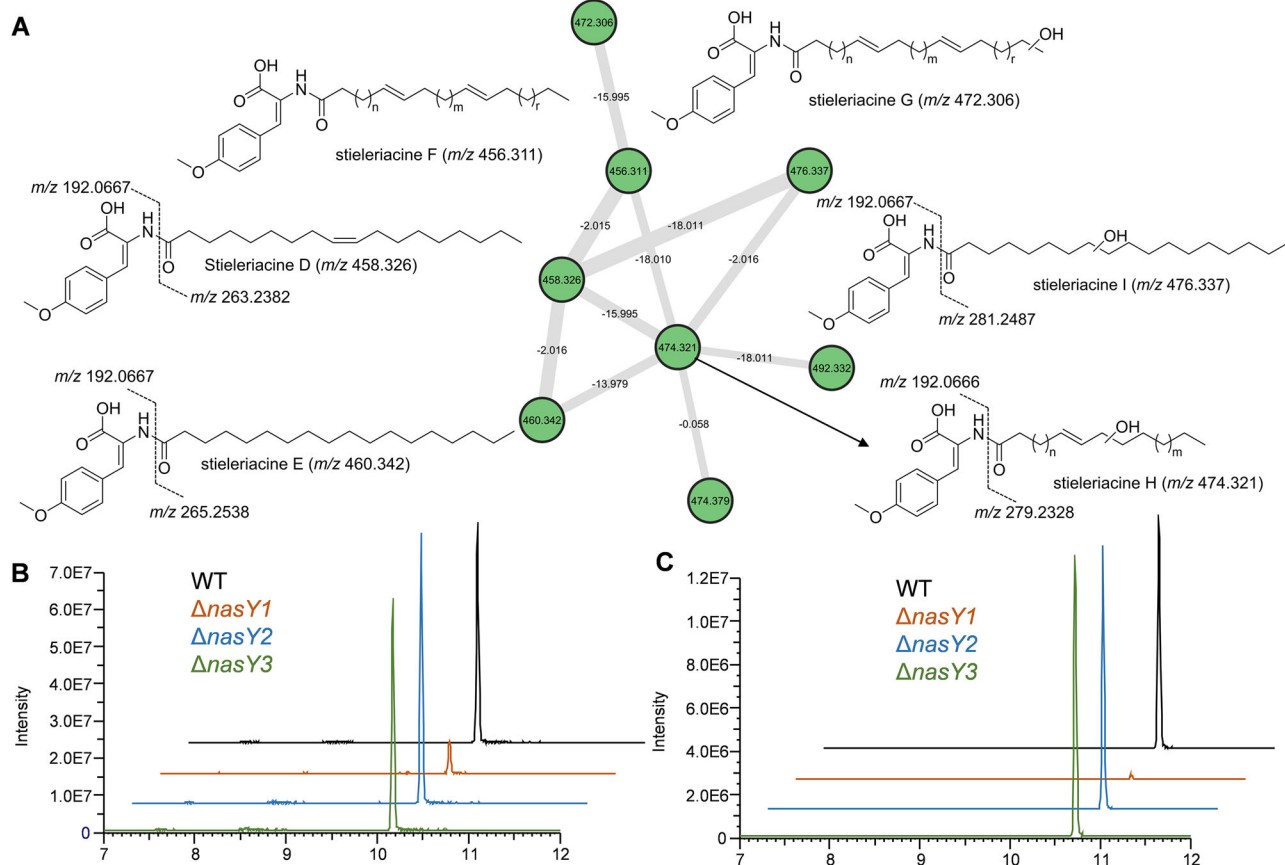

**Fig. 5 | Comparative analysis of *S. neptunia* Enr13ᵀ wild type and Δ*nasY1-3* knock-out mutants. A** Molecular network of stieleriacine-related molecular ion features, showing in addition to stieleriacine D and E, six additional mass features of stieleriacine derivatives. Structures of stieleriacine F, H, and I were predicted based on MS/MS fragmentation analysis. Extracted ion chromatograms of **B** stieleriacine D (*m/z* 458.326 [M + H]⁺) and **C** stieleriacine E (*m/z* 460.342 [M + H]⁺) of *S. neptunia* Enr13ᵀ wild type (WT; black), Δ*nasY1* (orange), Δ*nasY2* (blue) and Δ*nasY3* (green).

unwanted side reactions that could arise if starting directly from the amino acids. The compounds were acylated using three fatty acids: lauric acid, *E*-dodec-2-enoic acid, and *E*-hexadec-2-enoic acid for which two different coupling methods were applied. The first method followed a one-pot procedure described by Johansson et al.[31], utilizing *1,1'*-Carbonyldiimidazole (CDI) to activate lauric acid. The second method employed coupling reagents for activating unsaturated fatty acids during the coupling step. Using H-Tyr-OMe*HCl as the starting material afforded amides **18–20** in 60–91% yield, whereas H-Tyr(Me)-OMe*HCl gave amides **24–26** in 82–94% yield. Employing H-Tyr(3-Me)-OMe*HCl furnished amides **31–33** in 72–94% yield.

The resulting amides were directly subjected to saponification, achieving nearly quantitative yields of **21–23** and **27–29**, while only moderate yields were obtained for **33–35** for yet unknown reasons. However, any attempt to synthesize dehydrotyrosine-derivatives including e.g. stieleriacines A and B from these and other precursors failed despite several different synthetic procedures tested[32]. Overall, the synthesis of 21 *N*-acyl amino acids derivatives was achieved, including a stieleriacine C stereoisomer (**34**).

Comparison of the NMR data for isolated stieleriacine C with the synthetic product **34** and related derivatives revealed strong agreement in chemical shifts (¹H, ¹³C) and coupling patterns, confirming the identity of the isolated compound (Supplementary Table 2). However, a significant discrepancy was observed in the optical rotation values between isolated stieleriacine C ($[\alpha]_D^{21} = -28$ (c = 1 in MeOH)) and the synthesized compound ($[\alpha]_D^{20} = +28.3$ (c = 0.15 in MeOH)). This mismatch suggests an inverted stereocenter at C-2 for the natural product, and the synthesis of *epi*-stieleriacine C. Comparison of the analytical data of kyonggic acid C with

synthetic product **18** revealed strong agreement in chemical shifts (¹H, ¹³C) and coupling patterns, confirming the identity of the isolated compound. In addition, comparison of optical rotation values between isolated kyonggic acid derivative **6** and **7** ($[\alpha]_D^{20} = +3°$) and structurally related synthetic compounds (**21–23**) (Fig. 6) including previously reported *N*-myristoyl L-tyrosine exhibited almost exclusively positive values. Thus, we concluded that the natural stieleriacine C is likely composed of a D-amino acid, while kyonggic acids are composed of L-amino acids. At this stage, it remains speculative whether stieleriacine C is derived from L- or D-amino acids. While L-amino acids are generally more abundant in the cellular environment, the biosynthesis of stieleriacine C would require a yet enigmatic additional epimerization step, for example through a reversible oxireductase-mediated transformation (Fig. 2), or alternatively the formation of a yet unidentified tyrazolone-like intermediate, as reported for *Mycobacterium tuberculosis* (Fig. 3).

**Bioactivity**

Previous studies have demonstrated that NAAs, including stieleriacines, exhibit a broad range of bioactivities including antimicrobial activity against human opportunistic bacterial pathogens, while *N*-acylated tyrosine derivatives have been shown to act as inhibitors of tyrosinase[20]. Selected compounds were tested for antimicrobial activity against a panel of test strains in a standardized disk diffusion assay (Supplementary Table S3). The results revealed that methyl ester derivatives (e.g. **18–20**, **24–26**) were generally inactive against the panel of test strain, while the respective acid derivatives (**21–23**, **27–29**) showed growth inhibitory activity against *Staphylococcus aureus* 134/94 (MRSA), *Enterococcus faecalis* 1528 (VRSA) and *Mycobacterium vaccae* 10670, although bacterial colonies were observed within

**Fig. 6 | Overview of the synthetic procedure towards *N*-acyl tyrosine derivatives. A** General reaction scheme. **B** Overview of obtained products and yields. **C** Comparative analysis of optical rotations of isolated stieleriacine C, synthetic *epi*-stieleriacine C and kyonggic acids.

the inhibition zones indicating emergence of resistance. It is worth noting that the presence of a *para*-methoxy group or a free *para*-hydroxy group had no significant effect on the antimicrobial properties.

## Conclusion

The phylum *Planctomycetota* still poses significant challenges for targeted genome mining efforts as functional biosynthetic components are often dispersed across the genome rather than clustered in canonical biosynthetic gene clusters. Thus, manual genome mining paired with targeted knock-out studies was essential to gain first insights into the biosynthesis of *N*-acyl tyrosine-derivatives: stieleriacines.

This study provides first concepts on the targeted knock-out of the major NAS-encoding gene of stieleriacine biosynthesis and verifies the predominantly responsible enzyme candidate for stieleriacines in *S. neptunia* Enr13[T]. Our molecular biological studies also pave the way for more in-depth molecular biological and biochemical studies of additional enzymes related to the biosynthesis of this compound class in *S. neptunia* Enr13[T] specifically, and more generally for the broader natural product family and members of this unexplored phylum *Planctomycetota*. Moreover, our synthetic approach enabled not only the synthesis of a dedicated compound library that can now be used as internal standards for quantification in future enzymatic and ecological assays but also the determination of the absolute structure of stieleriacine C and its unusual D-amino acid configuration through the synthesis of its epimer.

## Methods

### Genome analyses and construction of phylogenetic trees

GenBank- or RefSeq-annotated genomes of the analyzed producer strains were downloaded from NCBI (accession numbers provided in the results section) and BGCs were analyzed with antiSMASH v.8[33] In case that no NAS-encoding (*nasY*) gene was predicted by antiSMASH, the genome was annotated with eggnog-mapper v2.1.12[34] and the gene was identified by manual inspection of the re-annotated genome supported by protein blast analyses. For the construction of phylogenetic trees, protein sequences were aligned with ClustalW and trees were reconstructed with Fasttree 2.2 using the JTT + CAT model[35]. The trees were visualized with iTOL v6[36]. The

accession numbers of the used protein sequences are shown in brackets in the trees.

## Cultivation

*E. coli* TOP10 was used for cloning of plasmids harboring homology regions of *S. neptunia* and was cultivated in LB medium at 37 °C. Wild-type *S. neptunia* Enr13[T] and deletion mutants were cultivated in baffled flasks in M1H NAG ASW medium at 28 °C with shaking[37]. For cultivation of the deletion mutants, the medium was supplemented with 50 mg/L chloramphenicol, whereas 34 mg/L was used for *E. coli* during plasmid constructions.

## Construction of gene deletion mutants

Gene deletion mutants in *S. neptunia* Enr13[T] were constructed using a homologous recombination strategy enforcing two simultaneous crossing-over events based on a previously published protocol[38]. The expected deletion was checked by PCR-based amplification of the inserted region (chloramphenicol resistance gene) with primers binding outside of the up- and downstream homology regions used for the recombination.

## Extraction and UHPLC-ESI-HRMS analysis

After cultivation in three replicates, supernatant and biomass of *S. neptunia* Enr13[T] cultures were separated by centrifugation (4500 rpm, 4 °C, 30 min) (Supplementary Notes 1 and 2). The cell pellet was extracted with MeOH, while the culture supernatant was extracted with EtOAc. Organic fractions were combined and dried in vacuo. UHPLC-HESI-HRMS measurements of each sample were carried out on a Vanquish Flex UHPLC system (Thermo Scientific) combined with an Orbitrap Exploris 120 mass spectrometer (Thermo Scientific) equipped with a heated electrospray ionization (HESI) source. Metabolites were separated using reverse phase liquid chromatography at 40 °C using a Kinetex C18 column (50 × 2.1 mm, particle size 1.7 μm, 100 Å, Phenomenex) preceded by a C18 SecurityGuard™ ULTRA guard cartridge (2.1 mm, Phenomenex). Mobile phases consisted of $H_2O$ + 0.1% formic acid (buffer A) and acetonitrile + 0.1% formic acid (buffer B). Five μL sample concentrated at 200 μg/mL was injected into a gradient as follows: 0–1 min, 5% B; 1–10 min, 97% B; 10–12 min, 97% B; 12–13 min, 5% B; 13–15 min, 5% B at a constant flow rate of 0.3 mL/min. Data-dependent acquisition of MS2 spectra was performed in positive mode. MS1 full scans were recorded at *m/z* 150–1500 with a resolving power of 60,000 at *m/z* 200. Up to four MS2 spectra per MS1 survey scan were recorded with a resolving power of 30,000 at *m/z* 200 (see Supplementary Note 1 and 2).

## Molecular network analysis

A molecular network was created using the Global Natural Products Social Molecular Networking (GNPS) platform[30,39]. The precursor ion mass tolerance was set to 0.02 Da and an MS/MS fragment ion tolerance of 0.02 Da, while edges were filtered to have a cosine score above 0.8 and more than six peaks. Spectral networks were visualized using Cytoscape 3.10.2[40].

## Organic synthetic methods

For organic synthetic methods and analytical dataset, see Supplementary Note 3, Supplementary Table 2 and Supplementary Figs. 10–41).

## Reporting summary

Further information on research design is available in the Nature Portfolio Reporting Summary linked to this article.

## Data availability

HRMS/MS data have been deposited on the MassIVE server (MSV000098273; https://doi.org/10.25345/C5F766K86). NMR files deposited on Zenodo (https://doi.org/10.5281/zenodo.15737373).

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

## Acknowledgements
The authors would like to thank Huijuan Guo for initial work on this study. This work was funded by the Deutsche Forschungsgemeinschaft (DFG, German Research Foundation) under Project-ID 239748522—CRC 1127 Chem-BioSys. C.B. greatly acknowledges funding from the European Union's Horizon 2020 research and innovation program (ERC Grant number: 802736, MORPHEUS).

## Author contributions
Ma.Sa. performed the synthesis of the NAAs and characterization, My.St. cultivated the strains and contributed to the genome analyses, S.B. performed the comparative analysis of knock-out strains, N.K. constructed the deletion mutants, analyzed the genomes for suitable BGCs and performed the comparative genomics, C.J. and C.B. supervised the study and have written the manuscript. All authors designed figures and contributed to the manuscript text and agree with the submitted version.

## Funding

## Competing interests
The authors declare no competing interests. Dr. Huijuan Guo, named in the acknowledgements, is a Senior Editor for Communications Chemistry, but was not involved in the editorial review of, or the decision to publish this article.
