## [Transparent Peer Review file · Communications Chemistry]

Investigations into the biosynthesis of stieleriactins and related N-acyl tyrosines by comparative genomics, knock-out studies and total synthesis of epi-stieleriactin C

Corresponding Author: Dr Christine Beemelmanns

Version 0:

Reviewer comments:

Reviewer #1

(Remarks to the Author)

This manuscript by Sauer et. al. reported the identification and verification of the biosynthetic gene clusters (BGCs) that are responsible for the biosynthesis of N-acyl dehydrotyrosine from the planctomycetal strains of the eponymous genus *Stieleria*. The authors first used genome mining to identify the putative BGCs from the producers that encoding N-acyl amino acid synthases, methyltransferases, and nitro/oxidoreductases. Then, by gene knock-out studies in the type strain *S. neptunia* Enr13T, the *nasY1* from one of the identified gene clusters was characterized to be responsible for the biosynthesis of stieleriactins. In addition, a series of N-acyl tyrosine derivatives were chemically synthesized, including epi-stieleriactin C, and the antimicrobial bioactivity of these compounds were evaluated. This study significantly advances our understanding of the biosynthesis of N-acyl dehydrotyrosines produced by the species in the phylum Planctomycetota, where the functional biosynthetic components are generally dispersed across the genome rather than clustered in canonical BGCs. The reviewer would recommend the acceptance of this manuscript for publication after minor revisions.

The following minor issues should be addressed:

1. In the section of "Synthesis of N-acyl tyrosine derivatives and epi-stieleriactin C", lines 271-273 and 280-281, the starting materials and the products are not consistent to the structures shown in Figure 6.
2. As mentioned in Lines 282-284, the authors failed to synthesize the dehydrotyrosine derivatives even following the reported synthetic procedures (ref. 28 in the maintext). Could the authors explain the reasons?
3. Since the D-tyrosine is commercial available, it would be better to synthesize stieleriactin C with a D-amino acid to fully characterize its structure.
4. There are three NASs encoded in the genome of *S. neptunia*. What's the functions of *NasY2* and *NasY3*? Whether these two enzyme are real N-acyl amino acid synthases?

Reviewer #2

(Remarks to the Author)

This is a well-conceived and thoughtfully executed study that provides important insights into an underexplored class of microbial lipids. It makes a meaningful contribution to the field of natural product biosynthesis and the chemistry of N-acyl amino acids. Below are several specific comments for the authors' consideration:

1) Gene redundancy in NAS function

The residual production of stieleriactins in the Δ *nasY1* mutant is attributed to potential genetic redundancy; however, this has not been experimentally verified. Constructing double or triple mutants (e.g., Δ *nasY1/2/3*), or performing heterologous expression of each *nasY* gene, would help clarify whether functional overlap exists among the different NAS candidates.

2) Stereochemistry of stieleriactin C

The authors report that natural stieleriactin C contains a D-amino acid, which is an intriguing and uncommon finding. This observation raises several follow-up questions:

Are other stieleriactins (A, B, D, E, etc.) also derived from D-tyrosine?

A brief discussion of how D-amino acids are biosynthesized in bacteria (e.g., via racemases or epimerases), and their occurrence in N-acyl tyrosines or related metabolites, would strengthen the interpretation and context of this finding.

3) Function of NasY2 and NasY3

Since NasY2 and NasY3 do not appear to contribute significantly to stieleriace biosynthesis, their functional roles remain unclear. Could they be involved in the synthesis of other N-acyl amino acids with different amino acid or fatty acid specificities? A short speculative discussion or additional data (e.g., from metabolomics or expression analysis) would be valuable.

4) Literature coverage on N-acyl amino acid biosynthesis

The references cited are somewhat limited in scope. Recent advances in enzymatic synthesis and whole-cell biotransformations for N-acyl amino acid production—particularly those using engineered enzymes or microbial hosts—should be cited to provide a more comprehensive overview of the field.

Reviewer #3

(Remarks to the Author)

The manuscript from Sauer et al. regarding the biosynthesis of epi-stieleriace C in *Stieleria neptunia*. The manuscript is well written. In particular, this reviewer finds that the Introduction of the manuscript details the background on these little known metabolites, the N-acyl tyrosines, and does a nice job of setting up the research. Scientifically, this reviewer finds nothing to critique about this work. The in silico genomic studies to identify novel enzyme(s) for the N-acyl (dehydro)tyrosines or phenylalanines is based on a reasonable proposed biosynthetic scheme (Fig. 2). The knockout experiments in *S. neptunia* showing that NasY1 is the primary enzyme responsible for stieleriace biosynthesis is consistent with the data. This work sets future, more detailed studies for the biosynthesis of N-acyl tyrosines and, potentially, other N-acyl amino acids.

There are a few minor issues that the authors should address.

Line 206 – “more” should be eliminated from “A more broader”

Line 231-234 are awkward and unclear. These lines need to be rewritten.

Line 284 – “partially not yet reported”. What does “partially” mean here. This is confusing.

Version 1:

Reviewer comments:

Reviewer #1

(Remarks to the Author)

Since the comments raised by the reviewer have been addressed, the reviewer would recommend the acceptance of the revised manuscript for publication.

Reviewer #2

(Remarks to the Author)

I have no further comments on the manuscript. Most of the questions have been well addressed

Point-to-Point-Response

Reviewer #1 (Remarks to the Author): This manuscript by Sauer et. al. reported the identification and verification of the biosynthetic gene clusters (BGCs) that are responsible for the biosynthesis of N-acyl dehydrotyrosine from the planctomycetal strains of the eponymous genus Stieleria. [...]. This study significantly advances our understanding of the biosynthesis of N-acyl dehydrotyrosines produced by the species in the phylum Planctomycetota, where the functional biosynthetic components are generally dispersed across the genome rather than clustered in canonical BGCs. The reviewer would recommend the acceptance of this manuscript for publication after minor revisions. The following minor issues should be addressed:	We greatly appreciate the reviewers time to review and comment on the manuscript. We have addressed the questions below and within the manuscript using track changes.
1. In the section of “Synthesis of N-acyl tyrosine derivatives and epi-stieleriaccine C”, lines 271-273 and 280-281, the starting materials and the products are not consistent to the structures shown in Figure 6.	We apologize for the mistake – indeed the naming of starting material was wrong and the two sentences have been revised accordingly.
2. As mentioned in Lines 282-284, the authors failed to synthesize the dehydrotyrosine derivatives even following the reported synthetic procedures (ref. 28 in the maintext). Could the authors explain the reasons?	We thank the reviewer for bringing this to our attention. We indeed followed the procedure described in Ref. 28, which is based on the thalassotalic acids study. That work similarly reported increasing challenges in product isolation with longer fatty acid chains. As azlactones are unstable on silica, the authors precipitated them directly. In our case, however, the reaction did not proceed beyond the azlactone formation step. In all attempts, no precipitated or soluble product was detectable by NMR. We also note that the additional methyl substituent on the aromatic ring may adversely affect the reaction, as consistently lower yields were observed for all acylated derivatives compared to the corresponding tyrosine analogues. 
3. Since the D-tyrosine is commercial available, it would be better to synthesize stieleriaceine C with a D-amino acid to fully characterize its structure.	Yes, D-tyrosine is commercially available and could be transformed using the same sequence. However, the NMR and HRMS data of both enantiomers are identical and would not provide any additional analytical information. The only distinguishing feature of the natural isolated compound is the optical rotation, which exhibits a negative value instead of the positive value observed for the L-amino acid-containing N-acyl variants as well as the structurally related kyonggic acids. Since the epimer differs at only one stereocenter, the optical rotation should, by definition, differ only in sign. Therefore, in the interest of resource efficiency, we did not pursue the additional synthesis of the natural variant as the isolated version was already reported. As indicated in an additional comment, we therefore would like to focus in future studies on the biosynthesis and whether or not the D-amino acid containing natural product is indeed the primary biosynthetic product after acylation or rather a product of several consecutive biosynthetic steps.
4. There are three NASs encoded in the genome of S. neptunia. What's the functions of NasY2 and NasY3? Whether these two enzyme are real N-acyl amino acid synthases?	The functions of NasY2 and NasY3 remain currently unknown. Among the three nasY genes, nasY1, which encodes the N-acyl amino acid synthase with a confirmed major role in stieleriaceine biosynthesis, is the only one located in proximity to genes encoding enzymes of the PEP-CTERM exosortase system. This system is responsible for exporting proteins involved in the formation of extracellular polymeric substances (EPS) and has previously been reported to be genomically linked to N-acyl amino acid synthases (DOI: 10.1128/JB.05426-11). In contrast, the genomic location of nasY2, adjacent to methyltransferase and nitroreductase genes, is likely not coincidental and may indicate a distinct functional context. The presence of three nasY genes in S. neptunia and S. maiorica suggests potential differences in the substrate spectra of the respective enzymes, a hypothesis that we intend to investigate in future studies.
Reviewer #2 (Remarks to the Author): This is a well-conceived and thoughtfully executed study that provides important insights into an	We deeply thank the reviewer for this very encouraging feedback.

underexplored class of microbial lipids. It makes a meaningful contribution to the field of natural product biosynthesis and the chemistry of N-acyl amino acids. Below are several specific comments for the authors' consideration:	
1) Gene redundancy in NAS function The residual production of stieleriactins in the ΔnasY1 mutant is attributed to potential genetic redundancy; however, this has not been experimentally verified. Constructing double or triple mutants (e.g., ΔnasY1/2/3), or performing heterologous expression of each nasY gene, would help clarify whether functional overlap exists among the different NAS candidates.	Yes, the construction of double and triple mutants would indeed be the logical next step to investigate potential overlaps in the substrate spectra of the three NasY enzymes. However, this is currently technically not feasible in S. neptunia. At present, only a single positive selection marker (chloramphenicol resistance) is available for this marine species, as it is intrinsically resistant to other commonly used antibiotic classes with established selection markers (kanamycin, spectinomycin, gentamycin, β-lactams), and the tetracycline resistance gene is not properly expressed. Alternative strategies that rely on recycling the selection marker via integration–excision protocols are also not applicable. The most common approach using the sacB gene from Bacillus subtilis cannot be employed because S. neptunia does not tolerate the high sucrose concentrations required for the excision step. In parallel, we attempted to delete the locus harboring the four nasY genes in Stieleria maiorica (Mal15_36340 to Mal15_37430, see Fig. 3), but the deletion mutant could not be obtained, likely because one of the additional genes within this locus is essential. We agree that heterologous expression is an excellent alternative approach. Indeed, we plan to investigate the NasY enzymes, along with the associated methyltransferase and nitroreductase, in a follow-up study. However, as the heterologous expression of proteins from Planctomycetota has its own intrinsic challenges, we do consider this work beyond the scope of this manuscript.
3) Function of NasY2 and NasY3 Since NasY2 and NasY3 do not appear to contribute significantly to stieleriactin biosynthesis, their functional roles remain unclear. Could they be involved in the synthesis of other N-acyl amino acids with different amino acid or fatty acid specificities? A short speculative discussion or additional data (e.g., from metabolomics or expression analysis) would be valuable.	We note that this important point was also raised by Reviewer 1 (Comment 4), and we kindly refer the reviewer to our detailed response provided above. In addition, we have re-examined our metabolomic data and again did not detect any other N-acyl amino acid congeners. Such compounds would have been readily identifiable through MS/MS-guided molecular networking, given their highly characteristic fragmentation patterns.

2) Stereochemistry of stieleriaccine C The authors report that natural stieleriaccine C contains a D-amino acid, which is an intriguing and uncommon finding. This observation raises several follow-up questions: Are other stieleriaccines (A, B, D, E, etc.) also derived from D-tyrosine? A brief discussion of how D-amino acids are biosynthesized in bacteria (e.g., via racemases or epimerases), and their occurrence in N-acyl tyrosines or related metabolites, would strengthen the interpretation and context of this finding.	We are very thankful for the comment and there are indeed follow-up questions to be discussed. Most stieleriaccines are dehydro-variants and therefore do not possess a stereocenter. Only stieleriaccine C and the related kyonggic acids are fully saturated derivatives. The kyonggic acids likely contain L-amino acids, as indicated by their positive optical rotations, which are consistent with the synthesized compound 34 and related analogues. We have now updated Figure 2 to include the additional possibility that the putative nitroreductase may act reversibly, or that a related enzyme could catalyze the reduction of the electron-rich, redox-active double bond in unsaturated stieleriaccines. In this scenario, stieleriaccines could originate from natural L-amino acids but undergo redox-mediated epimerization in the case of stieleriaccine C. We acknowledge that this remains speculative and will require further detailed investigation.
4) Literature coverage on N-acyl amino acid biosynthesis The references cited are somewhat limited in scope. Recent advances in enzymatic synthesis and whole-cell biotransformations for N-acyl amino acid production—particularly those using engineered enzymes or microbial hosts—should be cited to provide a more comprehensive overview of the field.	We apologize for not included the vast literature scope and have now include more literature references covering enzymatic synthesis and whole-cell biotransformations for N-acyl amino acid production. We have addressed the point in the manuscript
Reviewer #3 (Remarks to the Author): The manuscript from Sauer et al. regarding the biosynthesis of epi-stieleriaccine C in Stieleria neptunia. The manuscript is well written. In particular, this reviewer finds that the Introduction of the manuscript details the background on these little known metabolites, the N-acyl tyrosines, and does a nice job of setting up the research. Scientifically, this reviewer finds nothing to critique about this work. The in silico genomic studies to identify novel enzyme(s) for the N-acyl (dehydro)tyrosines or phenylalanines is based on a reasonable proposed biosynthetic scheme (Fig. 2). The knockout experiments in S. neptunia showing that NasY1 is the primary enzyme responsible for stieleriaccine biosynthesis is consistent with the data.	We are very thankful for the time to conduct the detailed review.

This work sets future, more detailed studies for the biosynthesis of N-acyl tyrosines and, potentially, other N-acyl amino acids. There are a few minor issues that the authors should address.	
Line 206 – “more” should be eliminated from “A more broader” Line 231-234 are awkward and unclear. These lines need to be rewritten. Line 284 – “partially not yet reported”. What does “partially” mean here. This is confusing.	 - done - rephrased line 231-234 to make the statement more precise - indeed, we have rephrased line 284 to avoid confusion
Editorial	
Please ensure that the following requirements are met, and that any other relevant checklists are completed and uploaded under the 'Related Manuscript file' type with the revised article. Chemical and biomolecular materials: Characterization of chemical and biomolecular materials NMR spectra: Requirements for the provision of NMR spectra Life sciences reporting summary: Reporting requirements for life sciences research In the event that your manuscript is accepted, we will provide detailed guidance on our journal policies and formatting. You may however wish to ensure that the manuscript broadly complies with our house style at this stage. See our style and formatting guide (https://www.nature.com/documents/commsj-phys-style-formatting-guide-accept.pdf) and checklist (https://www.nature.com/documents/commsj-phys-style-formatting-checklist-article.pdf) for reference.	HRMS/MS data has been deposited on the MassIVE server (MSV000098273; doi: 10.25345/C5F766K86). Organic synthetic methods: For organic synthetic methods and analytical dataset, see Supplemental Note 3 NMR files have been deposited on Zenodo (DOI 10.5281/zenodo.15737373)
DATA SOURCES: We strongly encourage authors to deposit all new data associated with the paper in a persistent repository where they can be freely and enduringly accessed. We recommend submitting the data to discipline-specific, community-recognized repositories, where possible and a list of recommended repositories is provided at http://www.nature.com/sdata/policies/repositories	
Data availability statements and data citations policy: All Communications Chemistry manuscripts must include a section titled "Data Availability" at	HRMS/MS data has been deposited on the MassIVE server (MSV000098273; doi: 10.25345/C5F766K86).

the end of the Methods section or main text (if no Methods). More information on this policy, and a list of examples, is available at <http://www.nature.com/authors/policies/data/data-availability-statements-data-citations.pdf>.

- Accession codes for deposited data
- Other unique identifiers (such as DOIs and hyperlinks for any other datasets)
- At a minimum, a statement confirming that all relevant data are available from the authors
- If applicable, a statement regarding data available with restrictions
- If a dataset has a Digital Object Identifier (DOI) as its unique identifier, we strongly encourage including this in the Reference list and citing the dataset in the Data Availability Statement.

Organic synthetic methods: For organic synthetic methods and analytical dataset, see Supplemental Note 3

NMR files deposited on Zenodo (DOI 10.5281/zenodo.15737373)